# Space- and time-resolved small angle X-ray scattering to probe assembly of silver nanocrystal superlattices

Yixuan Yu [1], Dian Yu[2], Babak Sadigh[1] & Christine A. Orme [1]

The structure of nanocrystal superlattices has been extensively studied and well documented, however, their assembly process is poorly understood. In this work, we demonstrate an in situ space- and time-resolved small angle X-ray scattering measurement that we use to probe the assembly of silver nanocrystal superlattices driven by electric fields. The electric field creates a nanocrystal flux to the surface, providing a systematic means to vary the nanocrystal concentration near the electrode and thereby to initiate nucleation and growth of super-lattices in several minutes. Using this approach, we measure the space- and time-resolved concentration and polydispersity gradients during deposition and show how they affect the superlattice constant and degree of order. We find that the field induces a size-selection effect that can reduce the polydispersity near the substrate by 21% leading to better quality crystals and resulting in field strength-dependent superlattice lattice constants.

[1] Lawrence Livermore National Laboratory, 7000 East Avenue, Livermore, CA 94550, USA. [2] University of California, Los Angeles, 410 Westwood Plaza, Los Angeles, CA 90095, USA. Correspondence and requests for materials should be addressed to C.A.O. (email: orme1@llnl.gov)

Our society is in the information age characterized by the rising needs for large-area, low-cost, and flexible electronic devices[1–3]. Colloidal nanocrystals are promising building blocks for these devices, due to their size-tunable properties[4–7] and ability to be integrated into devices with low-cost solution processes[8–12]. Nanocrystals inside devices form ensembles, whose collective physical properties, e.g., charge carrier mobility[13,14], depend on both the properties of individual nanocrystals and the way they are arranged[15–18]. In principle, ordered nanocrystal ensembles, or superlattices[19–23], allow more controllable charge carrier transport by facilitating the formation of minibands[24–26]. In practice, however, few devices built from ordered nanocrystal superlattices have been reported. This situation can be improved if detailed quantitative information on the nanocrystal assembly process could be obtained and if the crystallization process were better controlled.

Recent advances in in situ techniques at synchrotron beamlines make it possible to probe nanocrystal self-assembly processes in controlled environments[27–30]. A series of interesting phenomena, including superlattice structural rearrangement[31–34], nanocrystal tilting and fusing[35], and reversible disorder–order transitions[36,37], have been observed. However, with a few exceptions[28], previous studies focused on the structure and structural transition of the superlattices themselves and little is learned about the assembly process that generates the superlattices, such as the evolution of nanocrystal solution concentration prior to and during the superlattice formation, and nanocrystal flux toward the substrate. This is likely because most previous studies use solution evaporation methods to generate nanocrystal superlattices and probe the assembly process as the solvent is being gradually removed[31, 38–41]. The fact that the volume and shape of the nanocrystal solution is continually changing in an uncontrollable manner and the fact that capillary forces can drive nanocrystal motion during drying, makes it difficult to obtain quantitative information on the assembly process. Electric field driven growth offers a solution to this problem.

Progress in electrophoresis and electrophoretic deposition has shown that electric fields can be used to drive nanocrystal motion for purification and film formation. For example, using gel or capillary electrophoresis researchers have separated nanocrystals as a function of size or shape or charge[42–44]. Similarly, electrophoretic deposition has been used to deposit nanocrystal films of metals, oxides, and semiconductors with a few researchers focusing on monodisperse nanocrystals that have the potential to form superlattices[45, 46]. However, until recently electrophoretic deposition of nanocrystals has generated films that do not have long-range 3D order, which has limited systematic study of the assembly process. We have recently demonstrated that an electric field can be used to drive the assembly of well-ordered, 3D nanocrystal superlattices[47]. Because the electric field increases the local concentration without changing the volume, shape, or composition of nanocrystal solution, the crystallizing system can be probed quantitatively without complications associated with capillary forces or scattering from drying interfaces.

In the present work, we take advantage of field driven assembly to simultaneously probe the superlattice and its fluid environment while superlattices nucleate and grow. By scanning a finely focused X-ray beam, we use space- and time-resolved small angle X-ray scattering (SAXS) to measure nanocrystal concentration and polydispersity both at the anode interface, where superlattices nucleate, and also throughout the fluid cell. From the space and time dependent concentration profiles we calculate the nanocrystal flux toward the surface and show that it is a function of the applied field strength. Once calibrated by these measurements, the applied field provides a systematically means to control the flux much as is done for vapor growth methods such as sputter deposition or molecular beam epitaxy. To demonstrate the advantages of flux control, we show that it can be used to systematically vary the nucleation density, which is an important strategy for controlling the average grain size or microstructure of a polycrystalline film. A serendipitous advantage of using electric fields to assemble nanocrystals is that the field creates a nanocrystal size gradient near the anode. Our polydisperity profiles show that the field focuses the nanocrystal size distribution near the anode by preferentially accumulating nanocrystals of larger size. This size-selection effect can reduce the polydispersity near the substrate by 21% leading to better quality crystals and resulting in field strength-dependent superlattice lattice constants.

## Results

**Electric field-driven assembly of nanocrystal superlattices.** 1-dodecanethiol capped silver (Ag) nanocrystals are chosen to be the model material because they are widely used building blocks for superlattices, and more importantly, they bear negative charges due to surface adsorption of bromide anions ($Br^-$) during synthesis, facilitating electric field-driven assembly[47]. The applied electric field drives negatively charged nanocrystals dispersed in anhydrous toluene to the anode (positively charged electrode), where they become neutralized and then accumulate. Ag nanocrystals with adsorbed $Br^-$ are neutralized at the anode likely by oxidizing $Br^-$ to $Br_2$. When the local nanocrystal concentration exceeds their solubility, faceted 3D superlattices nucleate and grow on the substrate (Supplementary Note 1). Figure 1a shows an optical microscope image of an anode under a 20 V cm$^{-1}$ field for 60 min, in which equilateral triangular islands are clearly present. Atomic force microscopy (AFM, Fig. 1b) indicates that these islands are ~1.3 μm thick. Figure 1c shows a representative scanning electron microscopy (SEM) image of an island with base and top edge lengths of 28.5 and 26.9 μm, respectively. A SEM image zoomed-in to the top of the island allows a view of the hexagonal packing of nanocrystals (Fig. 1d). Grazing incidence small angle X-ray scattering (GISAXS, Fig. 1e), probing the structure for a large number of islands, reveals that the superlattices have a face centered cubic (FCC) structure with a lattice constant of 13.2 nm, oriented with their (111) planes parallel to the substrate. The superlattices exhibit a high degree of order, as the diffraction spots associated with high Miller index planes, e.g. (228), (446), and (157), are clearly visible in the GISAXS pattern. Additional GISAXS, AFM, and optical data for other electric field strengths are shown in Supplementary Figs 1–3.

3D nanocrystal superlattices shown in Fig. 1a, although some of them are partially merged, have a uniform equilateral triangular shape, suggesting that they are single crystalline, consistent with the high degree of order observed via GISAXS. A single crystal FCC superlattice tends to expose the most densely packed {111} surfaces. As illustrated in the scheme in the bottom left of Fig. 1c, the observed shape is consistent with a tetrahedron with the top corner truncated. Such a truncated tetrahedron exposes five {111} planes and has a film thickness ($T_f$) depending on the length difference between the top ($L_{top}$) and base edges ($L_{base}$), $T_f = \sqrt{2/3}\left(L_{base} - L_{top}\right)$. For example, the superlattice shown in the Fig. 1c should have a thickness of $T_f = \sqrt{2/3}(28.5 - 26.9)\mu m = 1.3\,\mu m$, consistent with AFM results.

The nucleation density of superlattices, estimated by counting the number of superlattices on the substrate, is dependent on the applied electric field strength. For instance, Fig. 2a shows an SEM image for the sample made under a 20 V cm$^{-1}$ field, which exhibits a lower nucleation density than that made under a 120 V cm$^{-1}$ field (Fig. 2b). Superlattice nucleation densities, averaged

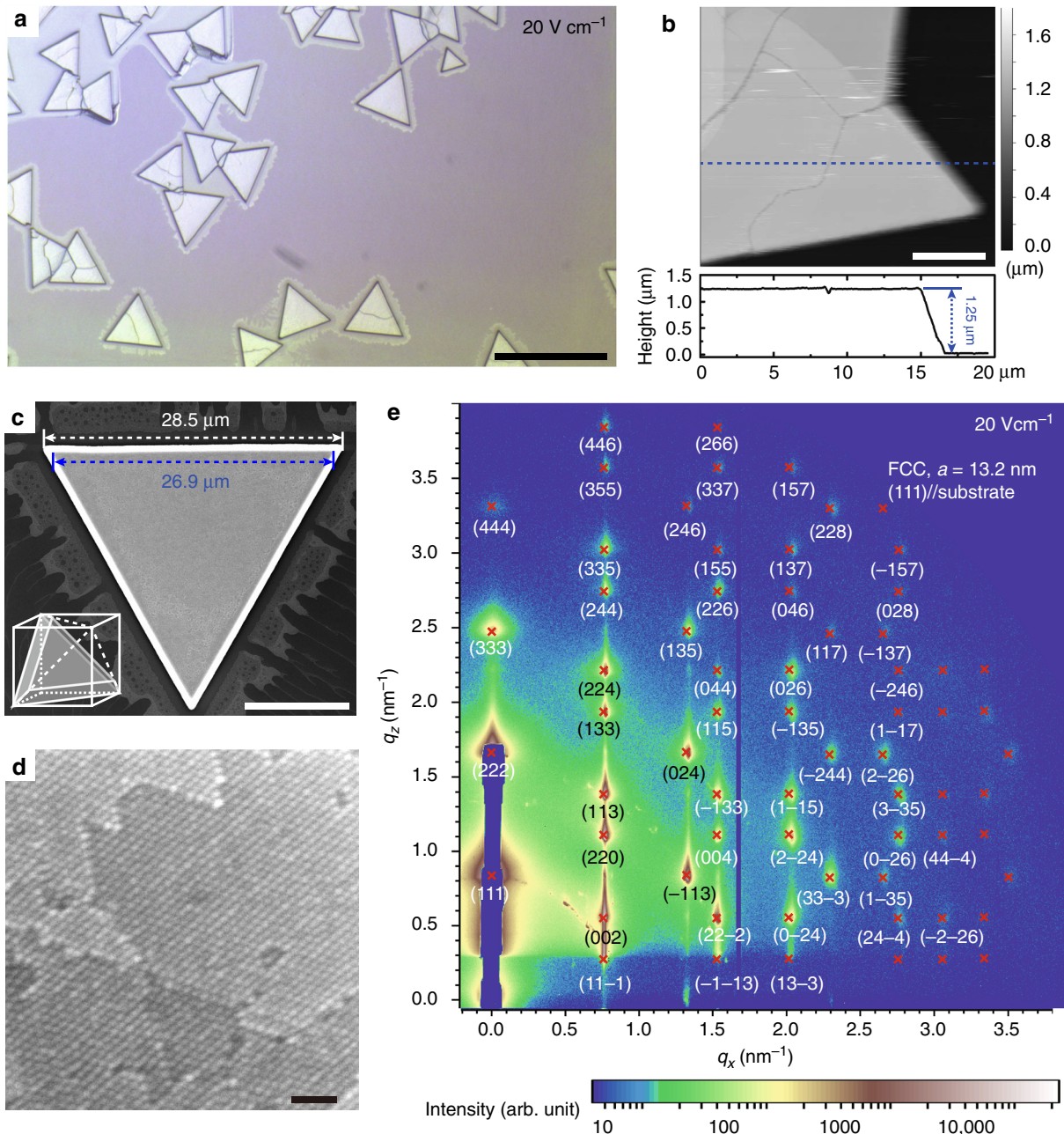

**Fig. 1** Electric fields assemble Ag nanocrystals into three-dimensional FCC superlattices. **a** Optical microscope image of three-dimensional (3D) superlattices grown under an electric field of 20 V cm$^{-1}$ for 60 min. These superlattices have a uniform equilateral triangular shape. Scale bar is 50 μm. **b** Atomic force microscope image with 5 μm scale bar of superlattices (top), and the height profile along the blue dotted line (bottom). **c** SEM image with 10 μm scale bar of a superlattice that has a base edge length of 28.5 μm (white dotted line) and top edge length of 26.9 μm (blue dotted line). Scheme inserted at bottom left corner shows the proposed geometry of a 3D superlattice, which is a truncated tetrahedron. **d** SEM image with 50 nm scale bar acquired at the top of a superlattice showing the packing of individual Ag nanocrystals. **e** GISAXS pattern of superlattices obtained under a field of 20 V cm$^{-1}$ for 60 min, indexed (red crosses) to an FCC structure with a lattice constant, $a$, of 13.2 nm, oriented with its (111) planes parallel to the substrate. The axes, $q_x$ and $q_z$ represent the magnitude of the scatter vector in the plane of the sample and perpendicular to the plane of the sample, respectively

over multiple SEM images (>5), are summarized in Fig. 2c. More details on counting islands after they merge are described in Supplementary Notes 2 and 3.

**In situ space and time-resolved small angle X-ray scattering.** Electric fields drive negatively charged nanocrystals to the anode and create a concentration gradient decaying from the anode surface to the bulk solution. We probed the evolution of the concentration gradient in real time with a spatial resolution of 50 μm using SAXS. As depicted in Fig. 3a, Ag nanocrystal solution (anhydrous toluene) is sealed in the liquid chamber and a vertical electric field is generated between two parallel, horizontally placed electrodes, with the anode on the bottom (see Supplementary Methods for details). The distance between the top and bottom electrodes is 3.5 mm and the thickness of the liquid cell is 10 mm. A finely focused X-ray beam, with an energy of 13.3 keV and full width at half maximum (FWHM) of 34 μm, is transmitted

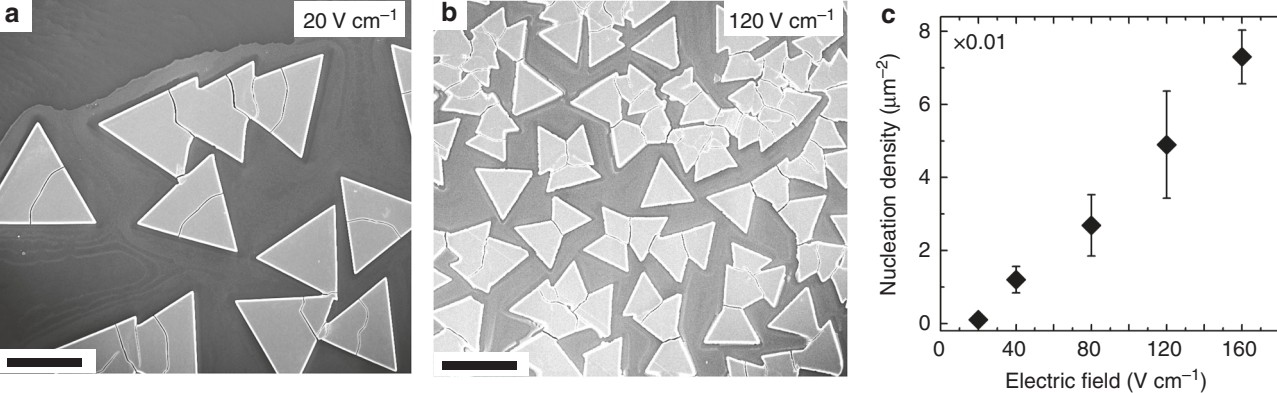

**Fig. 2** Superlattices exhibit higher nucleation densities under stronger electric fields. **a** SEM image of superlattices formed under 20 V cm$^{-1}$ electric field. **b** SEM image of superlattices formed under 120 V cm$^{-1}$ electric field. Both scale bars are 20 μm. **c** Nucleation density of superlattices plotted against the applied electric field strength. Error bars represent root mean square variations in the number of superlattice islands averaged over >5 SEM images

through the solution (10 mm thick), and the scattered photons are collected with a two-dimensional detector. In a typical experiment, the X-ray beam is first aligned with the surface of the anode and then scanned vertically from the anode surface to the bulk solution with a step size of 50 μm.

Figure 3b shows SAXS results for nanocrystal solution at various distances to the anode after applying a 57 V cm$^{-1}$ electric field for 55 min. Scattering intensity $I(q)$ is plotted at logarithmic scale against q-spacing in the left column. The SAXS intensity, $I(q)$, consists of form factor $P$, and structure factor $S$, $I(q) = P(q)S(q)$. The form factor reflects the size and shape of individual nanocrystals and the structure factor corresponds to the arrangement of nanocrystals in the ensemble[27, 30, 35, 36, 48]. In a dilute solution where the interaction between nanocrystals is negligible, the structure factor is a constant, whereas for ordered materials, the structure factor appears as sharp peaks in the scattering data. Similar scatter datasets are shown as waterfall plots for the four electric field strengths tested in Supplementary Figs. 4–7.

Prior to applying an electric field, SAXS data suggest a homogeneous nanocrystal solution absent of nanocrystal assembly (Supplementary Fig. 8). After applying a 57 V cm$^{-1}$ field for 55 min, a structure factor corresponding to an FCC structure appears in the SAXS data collected on the surface of anode ($z = 0$), while nanocrystals in the bulk solution remain un-assembled. The FCC superlattice has a lattice constant of 15.1 nm, which is substantially larger than the lattice constant of dried superlattices shown in Fig. 1, due to the presence of solvent molecules among nanocrystals. The mean core diameter of the nanocrystals is 7.1 nm, (Supplementary Fig. 8 and Supplementary Methods).

The same SAXS data are presented as Kratky plots[49], which are intensity multiplied by the square of the q-spacing ($q^2$) plotted against q-spacing (Fig. 3b, right column). The gray area underneath the Kratky plot, known as the SAXS invariant, $\int I(q)q^2 dq$, is proportional to the local volume fraction of nanocrystals[50]. Applying a field of 57 V cm$^{-1}$ has clearly caused the nanocrystals to accumulate near the anode, increasing the local nanocrystal concentration while at the same time depleting the concentration close to the cathode. Kratky plots of additional time points are shown in Supplementary Fig. 9.

The nanocrystal solution used in the SAXS experiments has an initial concentration of 4 mg/mL, corresponding to a volume fraction of 0.08% (see Supplementary Methods for details). By comparing the measured SAXS invariants to that of the initial solution of known volume fraction, we measure the volume fraction for the entire system in real time. Typical data can be

displayed as a 2D contour plot of the nanocrystal volume fraction, as a function of the distance to the anode and the time that the electric field has been applied for (Fig. 3c). Contour plots for additional electric field strengths are shown in Supplementary Fig. 10.

**Correlating superlattice growth with concentration and flux.** Nanocrystals assemble into superlattices on the anode as their local volume fraction rises. An advantage of in situ SAXS measurements is that both the structure factor and the invariant are measured simultaneously allowing a correlation between crystallization and solution concentration. Figure 4a shows the evolution of volume fraction profiles for nanocrystal solutions, plotted against the distance to the anode, while applying an electrical field of 57 V cm$^{-1}$. The nanocrystal volume fraction at the anode surface (where the nanocrystals assemble into superlattices) is plotted against time for various applied field strengths in Fig. 4b. Superlattices are found to form when the volume fraction rises to the range of 0.7–1.2%, the area highlighted with gray. No superlattice formation is observed during the experiment (~3 h) using a 14 V cm$^{-1}$ field, which never reaches this concentration range. It is worth noting that our experimental setup allows a spatial resolution of 50 μm, and the reported local volume fraction is a value averaged over this width. Figure 4a suggests that the volume fraction rises ~10 times within the 250 μm slice of solution next to the anode and rises even faster for solution closer to the anode. For this reason, we expect a significant volume fraction rise within the first 50 μm slice of solution. As a result, the volume fraction of nanocrystal solution in equilibrium with superlattices, which are 1–2 μm thick (as measured by AFM, Fig. 1b) should be much higher than the reported average value. A beam size of ~1 μm or smaller is needed to accurately measure the local volume fraction associated with the onset of crystallization in this system.

Electric fields drive negatively charged Ag nanocrystals toward the anode with a certain flux. The power of the space- and time-resolved SAXS techniques is that it allows us to quantify the nanocrystal flux from the volume fraction profile. For instance, the nanocrystal flux entering the solution within 50 μm of the anode can be calculated from the rate that the local volume fraction rises (see Supplementary Methods for details). Figure 4c shows the nanocrystal flux at near the anode under different applied electric field strength, indicating that stronger fields yield higher nanocrystal fluxes to the substrate, which consequently causes superlattices to nucleate at higher densities (Fig. 2c).

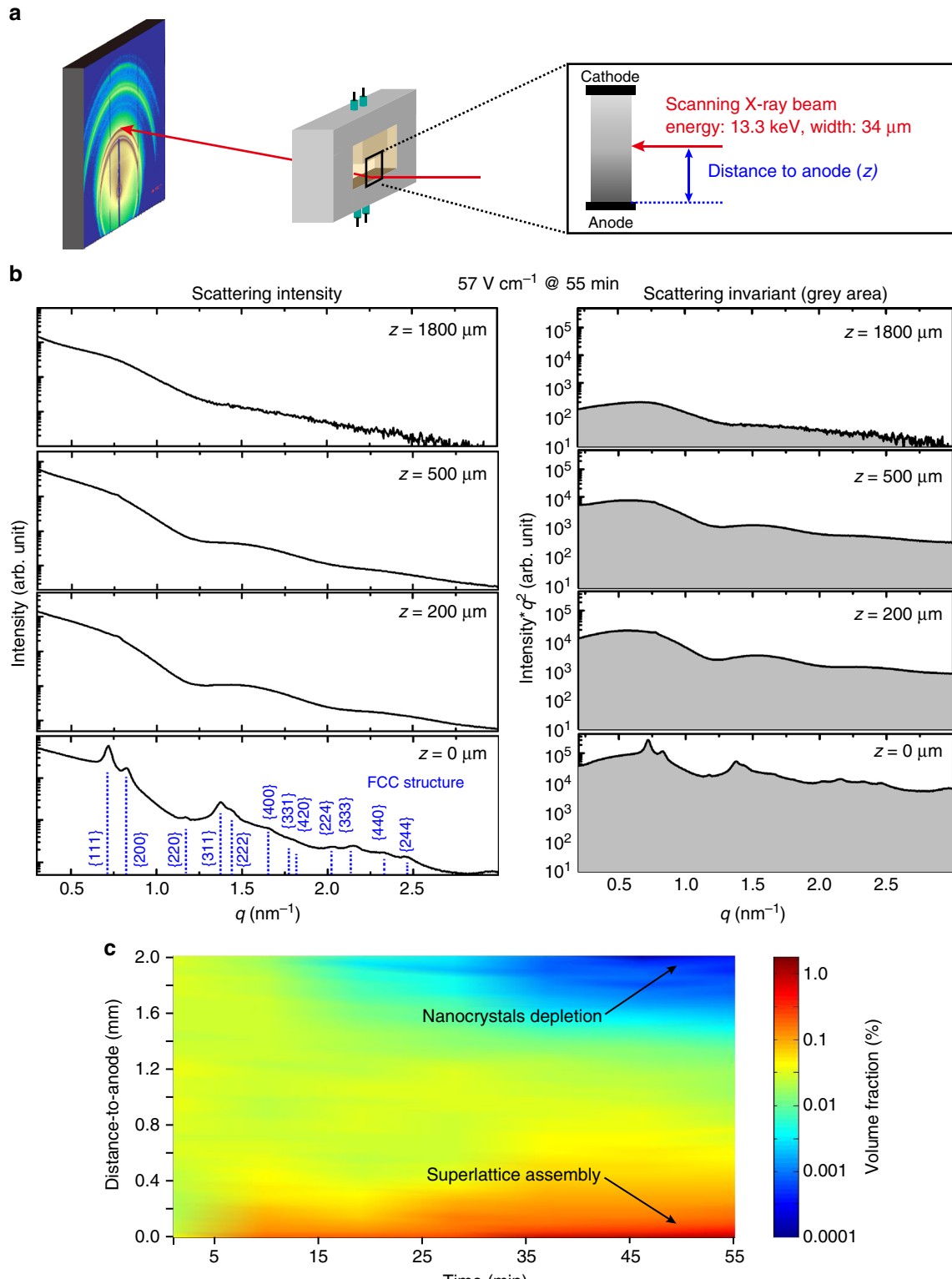

**Fig. 3** In situ space- and time-resolved SAXS to probe the electric field-driven assembly system. **a** Experimental setup of in situ SAXS experiments. **b** Left: scattering intensity, plotted on a logarithmic scale, for nanocrystal solution at various distances to the anode surface ($z = 1800$, 500, 200, and 0 μm, from top to bottom), after applying a field of 57 V cm$^{-1}$ for 55 min; and right: the corresponding Kratky plots that are displayed with the same scale to allow a direct visual comparison of invariant at difference locations. **c** Contour plot of the nanocrystal solution volume fraction against the distance to the anode and the duration of applied electrical field of 57 V cm$^{-1}$, obtained in a space- and time-resolved SAXS measurement. The color reflects the volume fraction % as shown in the scale at right

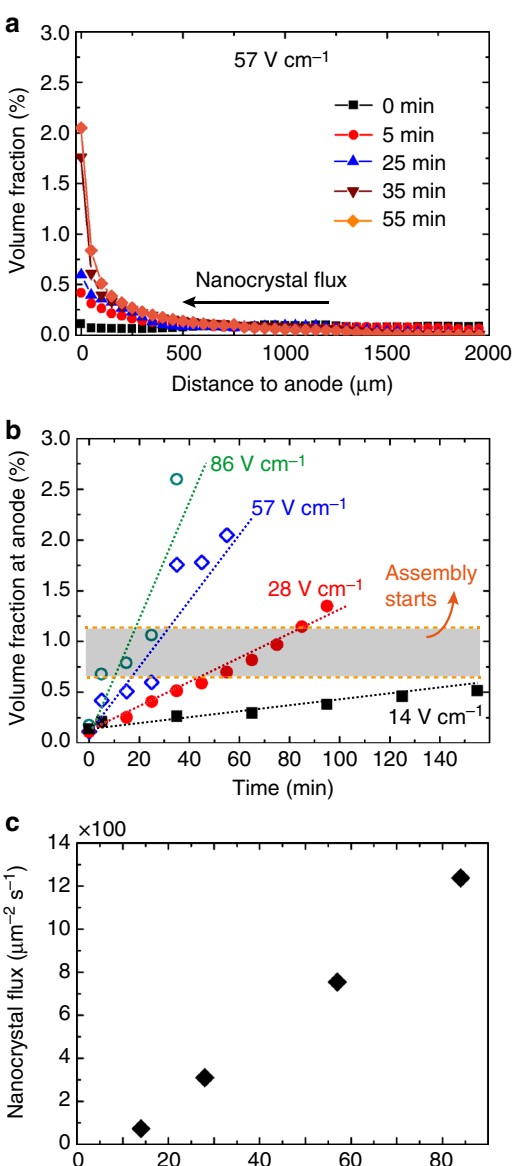

**Fig. 4** In situ SAXS shows higher nanocrystal flux to the anode under stronger fields. **a** Plots of volume fraction of nanocrystal solution against its distance to the anode, under an applied field of 57 V cm⁻¹ for various times. Black squares, red circles, blue triangles, brown upside down triangles, and orange diamonds are used for 0, 5, 25, 35, and 55 min, respectively. **b** The volume fraction of nanocrystal solution at the anode surface plotted against the time that electric fields have been applied for. Black squares, red solid circles, blue open diamonds, and green open circles represent fields of 14, 28, 57, and 86 V cm⁻¹, respectively. The dotted lines are linear fits. **c** The calculated nanocrystal flux entering the 50 μm thick slice of solution near the anode

**Electric field-induced size-selection of nanocrystals**. In the previous discussions, we have treated nanocrystals as if they are perfectly monodisperse. Real nanocrystal samples have size distributions that can be approximately treated as Gaussian distributions, $G(\bar{d}, \delta)$, in which $\bar{d}$ is the average diameter and $\delta$ is the standard deviation[32, 37]. The probability that a nanocrystal acquires a negative charge ($Q$) by adsorbing a Br⁻ anion is proportional to its surface area, $Q \propto d^2$. The electromagnetic force applied to this nanocrystal becomes $F_e = QE \propto d^2$, where $E$ represents the electric field strength. In steady state, the

electromagnetic force is balanced by a friction force, known as Stokes' drag, $F_f = 3\pi\eta d v$, where $\eta$ is dynamic viscosity of the solvent and $v$ is the nanocrystal drift velocity[51]. Therefore, a nanocrystal with diameter $d$ should have a velocity of

$$v = QE/3\pi\eta d \propto dE/3\eta \propto d, \qquad (1)$$

suggesting that larger charged nanocrystals migrate to the substrate faster than the smaller ones. In addition, the larger nanocrystals have smaller diffusion constants, $D = kT/3\pi d\eta$, where $k$ is the Boltzmann constant and $T$ is the temperature, suggesting that they diffuse away from the anode slower than the smaller ones once neutralized. This size-selection effect causes nanocrystals closer to the anode to have a larger average diameter, which is observed during in situ SAXS experiments, as we will show below.

Figure 5a shows Porod plots ($Iq^4$ against $q$) for a nanocrystal solution under a 57 V cm⁻¹ applied field for 55 min. Nanocrystal assembly is negligible in the solution away from the anode by more than 50 μm, and the scattering data are form factors of individual nanocrystals. The black open circles in the plots are the measured scattering data, and the blue curves are the best fits made by assuming a collection of non-interacting solid spheres with a Gaussian size distribution, $G(\bar{d}, \delta)$, where $\bar{d}$ represents the average diameter and $\delta$ its standard deviation, as described in Supplementary Methods[27, 37]. According to the fits, nanocrystals in the solution prior to applying electric fields have a diameter of 7.1 ± 1.1 nm ($\bar{d} \pm \delta$) with a polydispersity ($\delta/\bar{d}$) of 15.5% throughout the entire solution (Supplementary Fig. 8). After applying a field of 57 V cm⁻¹ for 55 min, nanocrystals 50, 500, 1000, and 1400 μm away from the anode have diameters and percent polydispersity of 7.4 ± 0.9 (12.2%), 6.9 ± 1.1 (15.9%), 6.5 ± 1.3 (20.0%), and 5.6±1.5 nm (26.8%), respectively. Larger nanocrystals have been preferentially accumulated near the anode and the size polydispersity has been reduced by 21%, compared to that of the original solution.

Figure 5b summarizes the average diameter (top) and size polydispersity (bottom) of nanocrystals in the final scan of each in situ experiment under various applied field strengths. The black dashed lines show the average diameter and polydispersity of the nanocrystal solution prior to applying fields. Electric fields distinguish nanocrystals of different size and alter their distribution in the solution. The field preferentially accumulates larger nanocrystals near the anode, focuses the local size distribution, and leaves behind nanocrystals that are smaller and of wider size distribution, thereby generating a size and polydispersity gradient. For all the studied field strengths, the average nanocrystal size at the anode surface are larger than the initial value prior to applying a field. The size increase, however, is less for a stronger field. This is because there are a finite number of nanocrystals in the system. Stronger fields drive more nanocrystals to the anode, which represents a larger portion of the total population and therefore has an average size closer to the entire population, resulting in a smaller change. In other words, the effect of selecting large nanocrystals is less pronounced for stronger fields. As a consequence of the size-selection effect, superlattices formed under weaker fields consist of nanocrystals of larger size, and therefore should have larger lattice constants. Figure 5c shows SAXS data collected at the anode surface in the final scan of each in situ experiment, in which diffraction peaks corresponding to FCC structures are clearly present. Diffraction peaks of superlattices formed under weaker fields appear at lower q-spacings, consistent with larger lattice constants.

Upon removal of the solution, superlattices formed on the substrate dry and contract. SAXS data for dried superlattices exhibit a similar trend for their lattice constant: larger lattice

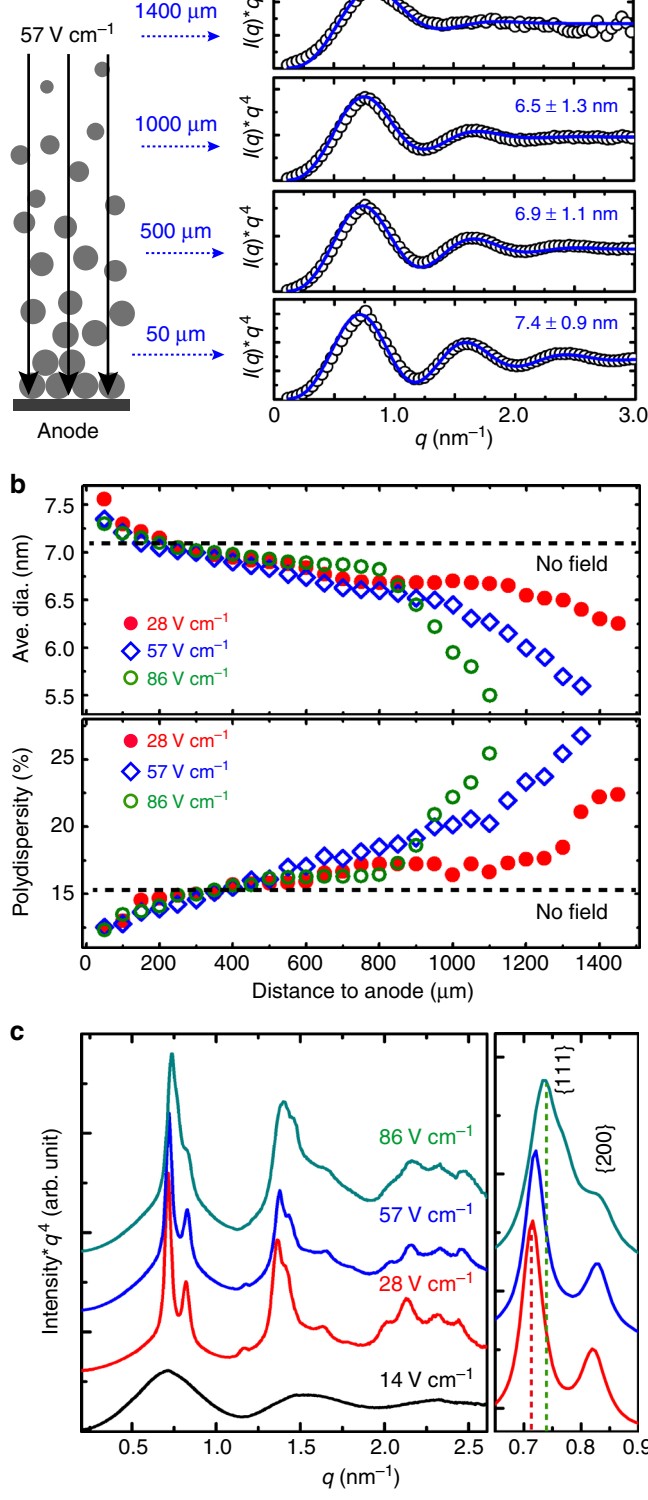

**Fig. 5** Electric fields cause nanocrystal size gradients leading to tunable lattice constants. **a** Porod plots of SAXS data for a nanocrystal solution under an applied field of 57 V cm$^{-1}$ for 55 min, acquired at various distances away from the anode. The black open circles are measured data, and the blue curves are the best fits with the average diameter and standard deviation indicated. The scheme at the left illustrates that the electric field (black lines) preferentially localizes larger nanocrystals near the anode, leading to a size gradient. **b** Average diameter (top) and polydispersity (bottom) of nanocrystals in the solution at various distances from the anode. Black dashed lines mark the value of original nanocrystal solution prior to applying fields. Red solid circles, blue diamonds, and green open circles show data for applied fields of 28, 57, and 86 V cm$^{-1}$, respectively. **c** In situ SAXS data of superlattices formed on the anode under various electric field strengths, while they are solvated in solution. Scatter data are plotted with black, red, blue, and green lines for applied fields of 14, 28, 57, and 86 V cm$^{-1}$, respectively

gradient that decays from the anode to bulk solution; this in turn causes the nanocrystals to diffuse back to the bulk solution. Overall, the electrical field-driven assembly process is an interplay between drift and diffusion of nanocrystals. Direct observation of this assembly process with in situ space- and time-resolved SAXS allows us to quantify the structure factor, the nanocrystal volume fraction, and the nanocrystal flux. Together these measurements correlate nucleation behavior with the surrounding solution environment, which is essential for understanding the crystallization process and connecting the experimental data with theoretical models.

Our results show that electric fields drive nanocrystals to the electrode providing a mechanism to regulate the flux. Flux control leads to tunable nucleation densities, which provides a handle on the domain size and hence film microstructure. In addition, the electric field preferentially accumulates larger nanocrystals at the substrate and focuses their size distribution, enabling the control of the superlattices lattice constant and lessening the requirement for size monodispersity. By coupling in situ space- and time-resolved SAXS with electric field driven assembly, we showcase a well-quantified, controllable path for making highly ordered 3D nanocrystal superlattices and a means to study the onset of nucleation.

## Methods

**Synthesis of Ag nanocrystals**. Ag nanocrystals are synthesized following a two-phase method reported in literature[48]. In a typical synthesis, 190 mg of silver nitrate dissolved in 30 mL of DI deionized water (aqueous phase) and 2.23 g of tetraoctylammonium bromide dissolved in 2.3 mL of chloroform (organic phase) are mixed and stirred at 400 rpm for 45 min, and the organic phase is extracted. Three hundred microliters of 1-dodecanthiol is added to the organic phase that is stirred for another 7 min. A total of 390 mg of sodium borohydride is dissolved in 30 mL of 4 °C DI water, and quickly pour into the organic phase. The two-phase mixture is stirred at 400 rpm for another 2.5 h before the organic phase, containing nanocrystals, is extracted. The organic phase is centrifuged at 8000 rpm for 5 min to precipitate poorly capped nanocrystals and/or their aggregates. The supernatant is mixed with equal-weight ethanol and centrifuged again at 8000 rpm. The nanocrystals aggregate and precipitate during the centrifugation and are re-dispersed in toluene at a concentration of 6 mg/mL. The nanocrystal solution is stored in a glass vial until further use. No size-selective precipitation is performed.

**Electric field-driven assembly of superlattices**. Ag nanocrystals superlattices are formed by applying an electric field between two parallel electrodes immersed in a nanocrystal solution (anhydrous toluene) at a concentration of 4 mg/mL. The electrodes are 12 mm × 25 mm double-side polished p-type silicon wafers, coated with gold on one side. In a typical superlattice assembly process, two electrodes are placed with uncoated silicon side facing each other at a distance of 2.5 mm, and a Keithley 2400 or 2450 source meter is used to apply a voltage between them, generating a uniform electric field. After applying the field for a certain time, the silicon wafer electrodes are withdrawn from the solution and dried in ambient

constants for weaker fields (Fig. 6a). The dependence of lattice constant on applied field strength of solvated and dried superlattices is summarized in Fig. 6b. The superlattice volume contracts by ~35% during drying.

## Discussion

Electrical fields cause electrophoretic forces that move negatively charged Ag nanocrystals to the anode, generating a concentration

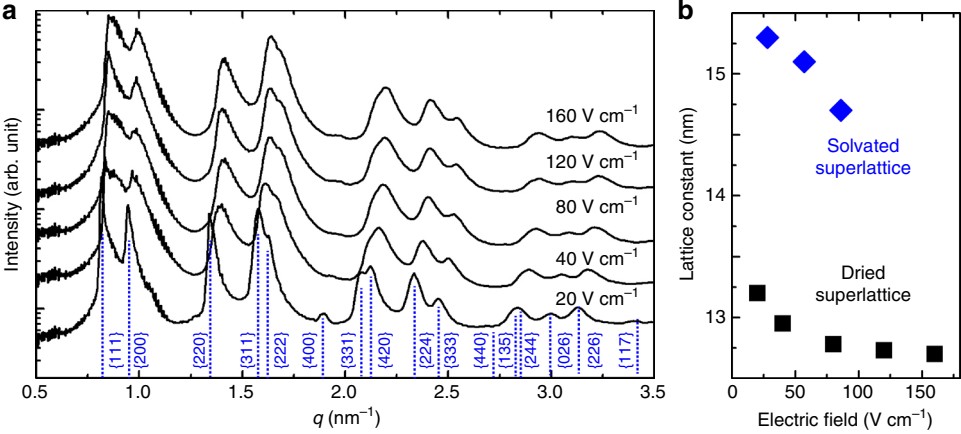

**Fig. 6** GISAXS pattern and lattice constant of dried nanocrystal superlattice. **a** Radial integration of GISAXS patterns of superlattices formed under various electric field strengths. **b** Lattice constants plotted against applied electric field strength for solvated (blue diamonds) and dried superlattices (black squares)

conditions for 5–10 s before the electric field is turned off. The superlattices are formed on the anodes that are positively charged during the assembly process.

**X-ray scattering measurements.** Ex situ GISAXS was performed at the Advanced Light Source (ALS) beamline 7.3.3 using 10 keV radiation with a beam spot size of 1 mm (horizontal) by 0.8 mm (vertical) at the sample position[52]. Scattering photons were collected with a Pilatus 2 M detector with a resolution of 1475 × 1679 pixels and a pixel size of 172 μm × 172 μm. The sample-to-detector distance was 1833.01 mm, the typical exposure time was 1–2 s, and the incident angle was 0.31°. The data were processed with Igor Pro (ver. 6.37, WaveMetrics, Inc) based software packages written by and available for download from Jan Ilavsky[53].

In situ space- and time-resolved SAXS was performed at the Advanced Photon Source (APS) beamline 12-ID-B with finely focused 13.3 keV radiation that has a full width at half maximum (FWHM) of 34 μm. The scattered photons were collected with a Pilatus 2 M detector that was 1993.89 mm away from the sample, calibrated with a silver behenate standard. The scattering patterns were radially integrated with a home-built MATLAB package, generated by Byeongdu Lee and is available by request to beamline users. The experimental apparatus was a customized cell shown in Supplementary Fig. 11. The cell contains top and bottom electrodes that were connected to external leads, and that allowed us to seal the nanocrystal solution inside the liquid chamber and transmit X-ray beam through the Kapton windows. The body of the cell was made from PEEK and the electrode was the same as described above except with a dimension of 10 mm × 25 mm. The distance between the electrodes was 3.5 mm and the thickness of the liquid chamber was 10 mm. Nanocrystal solution was injected through a pin hole on the top of the cell, which was sealed with a piece of parafilm tape during the experiment to minimize solvent evaporation.

## Data availability

The authors declare that all data supporting the findings of this study are available within the paper and its Supplementary Information Files. Electronic format of data is available from the corresponding author upon reasonable request.

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

## Acknowledgements

This work was supported by the Lawrence Livermore National Laboratory Directed Research and Development Program, 16-ERD-033. Work was performed under the auspices of the U.S. Department of Energy by Lawrence Livermore National Laboratory under Contract DE-AC52-07NA27344. We thank Scott Fisher and David Guerra for assistance with designing and building the in situ fluid cell. X-ray scattering experiments were performed at Advanced Light Source supported by the Office of Science, Office of Basic Energy Sciences, the U.S Department of Energy under contract no. DE-AC02-05CH11231, and Advanced Photon Source, a U.S. Department of Energy (DOE) Office of Science User Facility operated for the DOE Office of Science by Argonne National Laboratory under Contract No. DE-AC02-06CH11357.

## Author contributions

Y.Y. and C.A.O. conceived and designed the project. Y.Y. synthesized the materials, performed X-ray scattering experiments and analyzed the data. D.Y. performed AFM experiments. Y.Y. and D.Y. performed SEM experiments. B.S. aided in theoretical analysis. All authors discussed the results and interpretation. Y.Y. wrote the manuscript with contributions from the other authors.

## Additional information

**Competing interests:** The authors declare no competing interests.

