## [Peer Review File · Nature Communications]

Redactions:

Reviewers' comments:

Reviewer #1 (Remarks to the Author):

This well written and organized investigation of the assessment of order within the electric field-driven assembly of silver nanoparticles possesses omissions in its investigation or in its citations, which make me question the true novelty of this work.

1) Electric field assisted size selection and assembly of nanoparticles, particularly metallic nanoparticles, is not a new endeavor. Although the authors structures appear to have achieved order, forming such superlattices and realizing such size selection has been addressed in the following articles:

- a) Electrophoresis, Volume 34, Issue 6, March 2013, Pages 911-916.
- b) Nano Lett., 2007, 7 (9), pp 2881–2885
- c) Anal Bioanal Chem (2011) 399: 2831

2) Using x-rays to probe order within nanoparticle assemblies also is not a novel practice.

- a) ACS Nano, 2014, 8 (12), pp 12843–12850
- b) Nanoscale, 2014,6, 4047-4051

3) Measuring the velocity of nanoparticles in solution is not a novel endeavor. To my mind, determining the mobility or the zeta potential is the more typical practice when employing charged nanoparticle solutions for (self/electric field/magnetic field/evaporation assisted) assembly. Measuring the electrophoretic mobility or the zeta potential would have provided empirical information about the velocity of the nanoparticles in solution as they traveled from the region between the electrodes toward the positive electrode's surface, where they assembled. Perhaps, this is what the authors intended to communicate in their calculations and assessment; however, I am accustomed to see mobility or zeta potential data whenever a suspension or solution of nanoparticles/nanocrystals are involved. I did not find this information in the manuscript or in the supporting information, which seemed odd.

4) What other nanoparticle to nanoparticle forces were controlled to overcome Coulomb repulsion forces that the like-charged nanoparticles would have experienced during the assembly? Even in electrophoretic film formation situations, for which much larger electric fields are used (> 1kV/cm), electrostatic repulsion among the nanoparticles can give rise to weak ordering among the nanoparticles.

5) Stating that "...electric fields drive nanocrystals to the electrode providing a mechanism to regulate the flux," and that "...the electric field preferentially accumulates larger nanocrystals to the substrate and focuses their size distribution..." are common tenets in the electrophoresis and electrophoresis deposition communities and, hence, does not reveal new information to push forward the frontier of the community's knowledge.

Could the authors address these points?

Reviewer #2 (Remarks to the Author):

This papers presents interesting results on the assembly of silver colloidal nanocrystals under the effect of an electric field and the direct probing of the assembly by time and space resolved small

angle X-ray scattering.

Overall, I think the results are new, interesting and worth publishing in Nature Communication. However I have reservations on the structure of the paper and I feel the results could be presented in a much better way for the paper to be more convincing.

The paper is decomposed in several distinct parts, in the following I comment each part separately:

In the first one, the authors show how Ag nanocrystal superlattice are formed on the anode of their electro-chemical cell. The structure of the superlattice is revealed using various techniques such as GISAXS, AFM and SEM. These results have already been presented in a previous paper and set the stage for the following.

In the second part, the time-space resolved SAXS study is described. Using the invariant, the concentration of nanoparticles is measured as a function of time and depending on the distance from the electrode. The structure of the superlattice as probed with SAXS should be described in details in this part which is not the case. What is the cell parameter of the FCC lattice? How does the parameter compare with the mean size of the particles ?

Figure 2 is not satisfactory for me. The authors should show the SAXS patterns in I vs q in a log/log scale since it is the more common representation. On the 15 pannels of 2b, 12 look very similar. The authors should showcase more clearly the superlattice formation by showing a graph like figure 6f at this place ($I(q)$ at different z for a sufficiently long time so that SL have appeared).

In the third part, differential equation relating the flux to the diffusion coefficient. This part is incomplete to me since there should also be an equation relating the field to flux to the field and the charge of the particles...

I would not care if there were no model at all and just graphs showing that flux depend on the field but if the authors want to include a proper model, they should do it correctly.

figure 3e and f seem to be the same as figure 4a and 4b, please choose between the two, this is confusing.

Is it necessary to show all the volume fraction vs distance graphs for all the field values? Maybe just one (86 V/cm) is necessary since the information of the others is retrieved in figure 3e/f? Also a logcale in z would maybe be better?

The notation volume Fraction_ ϕ is weird. Choose between volume fraction and ϕ but the underscore is confusing.

The fourth part shows that size sorting occurs in the migration process since nanoparticle's migration velocity depend on their size. This is a very interesting finding ! This an important unambiguous proof that polydispersity affects the self-assembly into superlattice and that electric field can be used a size sorting mechanism. The authors should write clearly in the text the decrease in polydispersity that they can achieve. In the present form the figures in the text are size +/- poly in nm but do not display the % while it is in % in figure 5c. The two should be consistent !

The last part of the paper, on the field strength-dependent nucleation density is interesting but it is not clear why the authors chose to end the paper with it since it is not related to the previous section (size selection). It seems like coming back to the first part where the structure of the SL was

described.

One other general question and a remark:

One question raised by the paper is whether the superlattice nucleation is a bulk or a surface phenomenon. Do all the superlattice form at the surface of the electrode or are there some 3D superlattice which form in solution? The authors should be able to provide an answer to this question and discuss it. Are there any trace of structure factor far from the electrode?

As a point of curiosity, the charge regulation mechanism in organic solvent is known to be impacted by reverse micelles such as AOT. (see for example the recent work of Paul Bartlett at Bristol or the not so recent work by Dufresnes and Weitz in Langmuir). Have the authors look at the influence of reverse micelles on the assembly process? This of course could be the subject of future work and does not need to be included in the current paper.

Other more minor remarks:

- in the abstract, it is stated that the electric field assembly avoids "complications" of evaporative assembly. This is too vague. The authors should specify what they mean by complication
- still in the abstract: it is stated that the polydispersity is reduced by 4% near the substrate. In fact it is reduced by almost 30% since it goes from 15 to 11% if I understand well.
- the solvent should be precised at the beginning of the result section since it is important that all the experiments take place in an organic solvent.
- the thickness of the liquid chamber should be specified since it is an important parameter for SAXS experiment.
- usually the notation is $I(q) = P(q)S(q)$ with $P(q) = F^2(q)$ the form factor being sometimes F and sometimes P but the notation $I(q) = F(q)S(q)$ is rarely seen...

Reviewer #3 (Remarks to the Author):

In this work, the self-assembly of Ag nanocrystals into fcc crystalline solids is studied. More specifically: an electric field is applied between two electrodes driving the nanocrystals to the anode. At this place, the increased concentration initiates the formation of fcc superlattices. The entire process is monitored in situ by small-angle X-ray scattering. The authors discuss and display the assets of electric field driven assembly. Evaluation: This paper forms a thorough contribution to field-driven self-assembly of metallic nanoparticles into (conventional) fcc structures. Although it is not clear what the applications of metallic densely packed fcc NC solids could be, the fundamental aspects are thoroughly investigated and well presented in this ms. I therefore recommend publication in Nature Communications, after care is taken of the following comments.

Comments:

sub1: The authors show the effect of an electric field on the self assembly of "Charged" metallic nanocrystals. The authors state that the charging is due to adsorbed bromine ions. The electric polarization of the Ag NCs should also be important, in my opinion. I am not sure that the results for semiconductor NCs will be similar. Hence, the title, i.e. "nanocrystal superlattices" is too general and should focus on metallic NCs.

Sub2: The authors state that the electric field acts on the Br⁻ charges adsorbed on the nanocrystals. The origin of these charges, the average number density, and especially the way that the nanocrystals

become de-charged at the anode should be clarified. What happens to the absorbed Br⁻ ions at the anode?

Sub3: The measurement of the volume fraction of nucleation and growth is problematic due to the large spot volume. As the determination of NC concentration profiles is key, the authors should explain how they manage to obtain this profile in a seemingly quantitative way.

Sub4: Remarkably, the lowest fields give the largest superlattice domains, and the best quality crystals according to the structure factor profile, figure 6 f. Please comment, especially the paradox that with the lowest fields, the polydispersity in the volume where the superlattices grow is the largest.

Sub5: It is stated that larger fields favors the drift of the larger NCs to the growth area. This seems to be in contradiction with the fact that the fcc lattice constant is the smallest for the strongest fields (Figure 6g). Please explain.

We thank the reviewers for their time and constructive criticism. As a result of the comments we have restructured the paper, changed several figures, updated the references, and (hopefully) clarified the text. We have addressed each comment in detail below.

Reviewer #1 (Remarks to the Author):

This well written and organized investigation of the assessment of order within the electric field-driven assembly of silver nanoparticles possesses omissions in its investigation or in its citations, which make me question the true novelty of this work.

1) Electric field assisted size selection and assembly of nanoparticles, particularly metallic nanoparticles, is not a new endeavor. Although the authors structures appear to have achieved order, forming such superlattices and realizing such size selection has been addressed in the following articles:

- a) Electrophoresis, Volume 34, Issue 6, March 2013, Pages 911-916.
- b) Nano Lett., 2007, 7 (9), pp 2881–2885
- c) Anal Bioanal Chem (2011) 399: 2831

Response: Thanks for the comment. The reviewer has pointed out that the use of gel electrophoresis for size separation of nanocrystals needs to be properly referenced. We appreciate the help and agree that this was an omission. We have added the articles to our reference list. Our work is significantly different from this literature in that our goal is not purification per se. We view our work as a means to study and improve 3D superlattice assembly. We use the electric field to not only create a size gradient, but also to drive assembly of superlattices. Beyond this we use in situ methods to measure the polydispersity and solution concentration and monitor their effect on order/lattice constant/nucleation density - all of which are new endeavors.

We also agree that electric field assisted deposition of nanocrystals is well represented in the literature and that our wording and lack of references implies otherwise, which was not our intention. However, we are not aware of papers that have demonstrated ordering of 3-dimensional superlattices using electric fields. We have added a clarification and references references. *While other groups have demonstrated nanocrystal deposition using electric fields, they have created amorphous films^{44,45} rather than 3D superlattices, which has limited systematic study of the assembly process.*

2) Using x-rays to probe order within nanoparticle assemblies also is not a novel practice.

- a) ACS Nano, 2014, 8 (12), pp 12843–12850
- b) Nanoscale, 2014,6, 4047-4051

Response: The reviewer has a fair point. X-ray scattering, GISAXS, SAXS and WAXS, are standard if not the most commonly used method to characterize nanocrystal assemblies. We chose these techniques because we (and the community) consider them the gold standard for obtaining statistically-averaged information on the structure and form factors of nanocrystals and their assemblies. X-ray scattering at synchrotron beamlines has also been coupled with environment control sample stages to allow in situ time-resolved measurement for samples under heating/cooling, pressurization, and solvent annealing conditions. We have cited a number of papers that reflect this body of work (including a recent review). However, we feel that the reviewer is under valuing the importance of combining space and time resolved data.

We note that the first reference provides an example of spatially resolved structure using TSAXS. This work is quite interesting but it is not an in situ study. The authors use dried films and thus have no time resolution nor information regarding the solution concentration or nanocrystal poly-dispersity during growth. An in situ study that used TSAXS would, as the reviewer suggests, be an interesting addition to the literature.

The second reference illustrates why our methodology and conclusions differ from the current literature. The Dickerson group utilize in situ GISAXS to investigate deposition of nanocrystals using EPD. Their work differs from ours in several important respects. First, they do not show spatially-resolved data and for this reason do not have any quantitative data on the concentration or polydispersity profile near their electrode. Our first paper had the same limitation, which is why we went to a specialized small-spot beamline. Second, the authors conclude that ordering only occurs during drying (due to capillary forces) for their system. And, third, they have submonolayer growth not 3-dimensional superlattices. We have closely followed the work of Dickerson and Herman, who have been pioneers in the study of nanocrystal EPD, but we have not found published papers that use time and space resolved in situ xray scattering or that have demonstrated 3D superlattice formation using EPD. For these reasons we believe that our work, for the first time, introduces in situ space- and time-resolved measurements that reveal details of the 3D superlattice assembly process that cannot be measured with previous X-ray scattering techniques.

3) Measuring the velocity of nanoparticles in solution is not a novel endeavor. To my mind, determining the mobility or the zeta potential is the more typical practice when employing charged nanoparticle solutions for (self/electric field/magnetic field/evaporation assisted) assembly. Measuring the electrophoretic mobility or the zeta potential would have provided empirical information about the velocity of the nanoparticles in solution as they traveled from the region between the electrodes toward the positive electrode's surface, where they assembled. Perhaps, this is what the authors intended to communicate in their calculations and assessment; however, I am accustomed to see mobility or zeta potential data whenever a suspension or solution of nanoparticles/nanocrystals are involved. I did not find this information in the manuscript or in the supporting information, which seemed odd.

Response: We understand the reviewer's concern and agree Zeta potential is the typical characterization tool for this. In fact, we have collected Zeta potential data, please see Figure S30 in the SI of our previous paper: <https://pubs.acs.org/doi/suppl/10.1021/acs.nanolett.7b01323> [Redacted]. The black curve, which shows a charge centered at zero, is after cleaning to removing excess TOAB from the system (via repetitive cleaning/centrifugation cycles) and the red curve, which shows that the nanocrystals are negatively charged, is after TOAB is added back. We feel that these sample averaged velocimetry measurements of Zeta potential are too broad to be useful except in a qualitative way. This is why we tried to develop a method to measure drift velocity with X-ray scattering. However, as another reviewer pointed out, this calculation is incomplete and we decide to move the velocity discussion to the supplementary. Our ongoing modeling effort is focused on understanding the nanocrystal motion.

[Redacted]

4) What other nanoparticle to nanoparticle forces were controlled to overcome Coulomb repulsion forces that the like-charged nanoparticles would have experienced during the assembly? Even in electrophoretic film formation situations, for which much larger electric fields are used (> 1kV/cm), electrostatic repulsion among the nanoparticles can give rise to weak ordering among the nanoparticles.

Response: The reviewer is correct that no superlattice formation could happen if the nanocrystals remain charged at electrodes, due to the Coulombic repulsion. In this work, the charged nanocrystals are neutralized at the anode prior to the assembly. Our previous work “10.1021/acs.nanolett.7b01323” shows a control experiment where we observe no nanocrystal assembly when we did not allow charge neutralization. We have this statement in the text to clarify this point “*The applied electric field drives negatively charged nanocrystals, dispersed in anhydrous toluene, to the anode (positively charged electrode), where they become **neutralized and then accumulate.***”

Herman’s group also came to this conclusion for the deposition of CdS nanocrystals and our recent work shows that we can design superlattice patterns by blocking charge transfer on certain part of the substrate with non-conductive coatings..

5) Stating that “...electric fields drive nanocrystals to the electrode providing a mechanism to regulate the flux,” and that “...the electric field preferentially accumulates larger nanocrystals to the substrate and focuses their size distribution...” are common tenets in the electrophoresis and electrophoretic deposition communities and, hence, does not reveal new information to push forward the frontier of the community’s knowledge.

Response: The reviewer is perfectly right that these concepts are well-known for electrophoresis field. We did not intend to imply that electrophoresis as a means of size-separation or EPD of nanocrystals was in itself novel. What we believe is novel is: 1) using these tenets of electrophoresis and electrophoretic deposition to grow well-ordered 3D superlattices; 2) using flux control inherent in EPD to systematically vary the nucleation density of superlattice islands; 3) measuring the space and time resolved concentration and polydispersity gradients during deposition; 4) measuring the flux; and 5) demonstrating quantitatively how the flux and polydispersity effects the superlattice constant and degree of order. What we have achieved is to extend the concepts of the electrophoresis community to the more recent nanocrystal superlattice assembly field. We believe that bridging an emerging field of study with well-established fields is one of the most efficient ways to advance the emerging field.

We also note that while flux control is a tenet of electrophoretic deposition, flux is rarely measured because one needs to know the concentration gradient at the interface. Many groups measure film growth rate but to obtain flux from growth rate one needs to know the sticking coefficient and surface concentration which are not typically known. So we believe even for conventional EPD (which does not attempt 3D ordering) these space-time measurements are lacking and can aid quantitative modeling.

Reviewer #2 (Remarks to the Author):

This paper presents interesting results on the assembly of silver colloidal nanocrystals under the effect of an electric field and the direct probing of the assembly by time and space resolved small angle X-ray scattering. Overall, I think the results are new, interesting and worth publishing in Nature Communication. However, I have reservations on the structure of the paper and I feel the results could be presented in a much better way for the paper to be more convincing.

The paper is decomposed in several distinct parts, in the following I comment each part separately:

In the first one, the authors show how Ag nanocrystal superlattice are formed on the anode of their electro-chemical cell. The structure of the superlattice is revealed using various techniques such as GISAXS, AFM and SEM. These results have already been presented in a previous paper and set the stage for the following.

In the second part, the time-space resolved SAXS study is described. Using the invariant, the concentration of nanoparticles is measured as a function of time and depending on the distance from the electrode. The structure of the superlattice as probed with SAXS should be described in detail in this part which is not the case. What is the cell parameter of the FCC lattice? how does the parameter compare with the mean size of the particles?

Response: Thanks for the comment, this is a good point. We added the following sentences to the paragraph that describes the superlattice structure. *“The FCC superlattice has a lattice constant of 15.1 nm, which is substantially larger than the lattice constant of dried superlattice shown in Figure 1, due to the presence of solvent molecules between nanocrystals. The mean core diameter of the nanocrystals is 7.1 nm, as we will discuss in greater details later in this paper”*

Figure 2 is not satisfactory for me. The authors should show the SAXS patterns in I vs q in a log/log scale since it is the more common representation. On the 15 panels of 2b, 12 looks very similar. The authors should showcase more clearly the superlattice formation by showing a graph like figure 6f at this place ($I(q)$ at different z for a sufficiently long time so that SL have appeared).

Response: Thanks for the helpful suggestion. We have modified the Figure 2 to include the intensity VS q plot to show case the superlattice formation and adjust the text accordingly. Please see the text for the change we have made.

In the third part, differential equation relating the flux to the diffusion coefficient. This part is incomplete to me since there should also be an equation relating the field to flux to the field and the charge of the particles... I would not care if there were no model at all and just graphs showing that flux depend on the field but if the authors want to include a proper model, they should do it correctly.

Response: The drift velocity is a function of the electric field via $v(z) = \mu E(z)$ where μ is the electric mobility (this is equation 1 in the current paper version). This can then be substituted into $J(z, t) = c(z, t)v - D \frac{\partial c(z, t)}{\partial z}$ to obtain a relationship between flux and field. That said, our graphs plot flux and velocity against the *applied* electric field not the local electric field that appears in the differential equations. We have changed the text throughout to call the electric field the applied electric field. We believe our flux and velocity plots are correct but agree that the calculation is not complete (without knowledge of the local E-field). We moved the calculation to the supplementary and focus only on the describing and discussing the data we have in this part.

figure 3e and f seem to be the same as figure 4a and 4b, please choose between the two, this is confusing.

Is it necessary to show all the volume fraction vs distance graphs for all the field values? Maybe just one (86 V/cm) is necessary since the information of the others is retrieved in figure 3e/f? also a logscale in z would maybe be better ?

The notation volume Fraction_\phi is weird. Choose between volume fraction and \phi but the underscore is confusing.

Response: We deleted Figure 3a, 3b, and 3d, and moved 4a and 4b to the supplementary, removed underscore and used “volume fraction”. However, we choose to display volume fraction profile in linear scale because we want to emphasize the concentrated area with high volume fraction, not the diluted one.

The fourth part shows that size sorting occurs in the migration process since nanoparticle's migration velocity depend on their size. This is a very interesting finding ! This an important unambiguous proof that polydispersity affects the self-assembly into superlattice and that electric field can be used a size sorting mechanism. The authors should write clearly in the text the decrease in polydispersity that they can achieve. In the present form the figures in the text are size +/- poly in nm but do not display the % while it is in % in figure 5c. The two should be consistent !

Response: Thanks for the positive comment. We have added percentage to the “size +/-” and inserted “*the size polydispersity has been reduced by 21%, compared to the original solution*” to the discussion.

The last part of the paper, on the field strength-dependent nucleation density is interesting but it is not clear why the authors chose to end the paper with it since it is not related to the previous section (size selection). It seems like coming back to the first part where the structure of the SL was described.

Response: Thanks for the suggestion. We broke apart Figure 6 and moved the nucleation density earlier in the text as a new Figure 2.

One other general question and a remark:

One question raised by the paper is whether the superlattice nucleation is a bulk or a surface phenomenon. Do all the superlattice form at the surface of the electrode or are there some 3D

superlattice which form in solution? The authors should be able to provide an answer to this question and discuss it. Is there any trace of structure factor far from the electrode?

Response: The majority of the superlattices nucleate on the surface, however we have evidence for 3D superlattices that nucleate in the solution away from the surface that fall on the substrate. The surface nucleated superlattices have a preferential orientation with (111) plane parallel to the substrate, while solution nucleated superlattices have random orientation once they “land” on the substrate. We added an optical microscope image (Figure S17) in the supplementary to illustrate this point. The current setup with an X-ray beam width of 34 μm does not provide sufficient spatial resolution to probe solution nucleation because it is most likely to occur within the first step where the surface nucleation dominates the signal. Our recent results with a 10 μm wide coherent X-ray beam does show solution nucleation, and we will have a separate report focused on the discussion of solution nucleation and surface nucleation of superlattices.

As a point of curiosity, the charge regulation mechanism in organic solvent is known to be impacted by reverse micelles such as AOT. (see for example the recent work of Paul Bartlett at Bristol or the not so recent work by Dufresnes and Weitz in Langmuir). Have the authors look at the influence of reverse micelles on the assembly process? This of course could be the subject of future work and does not need to be included in the current paper.

Response: Thanks for the suggestion, this is a great point. We have at a point briefly looked into reverse micelles as a potential mechanism to create charges for nanocrystals. We decided to move on because our dynamic light scattering data do not show micelles and we have used anhydrous toluene as our solvent. However, our recent data also suggest that we could have less than one charge per particle, which makes us re-consider reverse micelle mechanism.

Other more minor remarks:

- in the abstract, it is stated that the electric field assembly avoids "complications" of evaporative assembly. This is too vague. The authors should specify what they mean by complication

We have clarified the text. *“Electric-field driven assembly avoids complications such as capillary forces and changes in volume associated with evaporative assembly...”*

- still in the abstract: it is stated that the polydispersity is reduced by 4% near the substrate. In fact it is reduced by almost 30% since it goes from 15 to 11% if i understand well.

Thanks, we have fixed this both in the introduction and in the text.

- the solvent should be precise at the beginning of the result section since it is important that all the experiments take place in an organic solvent.

Response: We have solvent information added to the second sentence of results *“The applied electric field drives negatively charged nanocrystals, dispersed in anhydrous toluene, to the anode (positively charged electrode), where they become neutralized and accumulate.”*

- the thickness of the liquid chamber should be specified since it is an important parameter for SAXS experiment.

Response: We added a sentence “*The distance between top and bottom electrodes is 3.5 mm, and the thickness of the liquid cell is 10 mm*” to describe the liquid cell dimensions.

- usually the notation is $I(q) = P(q)S(q)$ with $P(q)=F^2(q)$ the form factor being sometimes F and sometimes P but the notation $I(q)=F(q)S(q)$ is rarely seen...

Response: Thanks for the correction, we changed to $I(q) = P(q)S(q)$.

Reviewer #3 (Remarks to the Author):

In this work, the self-assembly of Ag nanocrystals into fcc crystalline solids is studied. More specifically, an electric field is applied between two electrodes driving the nanocrystals to the anode. At this place, the increased concentration initiates the formation of fcc superlattices. The entire process is monitored in situ by small-angle X-ray scattering. The authors discuss and display the assets of electric field driven assembly. Evaluation: This paper forms a thorough contribution to field-driven self-assembly of metallic nanoparticles into (conventional) fcc structures. Although it is not clear what the applications of metallic densely packed fcc NC solids could be, the fundamental aspects are thoroughly investigated and well presented in this ms. I therefore recommend publication in Nature Communications, after care is taken of the following comments.

Comments:

sub1: The authors show the effect of an electric field on the self assembly of "Charged" metallic nanocrystals. The authors state that the charging is due to adsorbed bromine ions. The electric polarization of the Ag NCs should also be important, in my opinion. I am not sure that the results for semiconductor NCs will be similar. Hence, the title, i.e. "nanocrystal superlattices" is too general and should focus on metallic NCs.

Response: The reviewer has a good point. We changed the title to “*In Situ Space- and Time-Resolved Small Angle X-ray Scattering to Probe Electric Field-Driven Assembly of Silver Nanocrystal Superlattices*”, since Ag nanocrystal is all we have studied in this work.

But to address the comment, we believe that our particle motion is dominated by charge rather than polarization for a couple reasons. The particles only deposit substantially on the positive electrode. If we were dominated by polarization then the particles should be equally attracted to both electrodes. Also, our recent work with semiconducting nanocrystals, suggests the results could be similar if the surface chemistry is tuned properly.

Sub2: The authors state that the electric field acts on the Br⁻ charges adsorbed on the nanocrystals. The origin of these charges, the average number density, and especially the way that the nanocrystals become de-charged at the anode should be clarified. What happens to the absorbed Br⁻ ions at the anode?

Response: The reviewer has a really good point. We came to the conclusion that origin of negative is Br⁻ adsorption from our previous work [10.1021/acs.nanolett.7b01323](https://doi.org/10.1021/acs.nanolett.7b01323) (referenced). In that work, we start with pure Ag nanocrystals and found they are positively charged, we then introduced Br⁻ ions (adding TOABr) and flipped the nanocrystal charge from positive to negative. We have to admit that we are not completely clear on the average number of Br⁻ on nanocrystal, as our attempts to collect conclusive Zeta potential results has yet to be successful, due to wide Zeta potential distribution. We propose that Br⁻ is oxidized to Br₂ at anode, however, do not have conclusive data in hand to prove this. Our ongoing work is focused on studying the trace product of nanocrystal neutralization. We agree that a description or hypothesis for how nanocrystals are neutralized should be included in the discussion, and we added this sentence to the first paragraph of results “*Ag nanocrystals with adsorbed Br⁻ are likely neutralized at the anode by having Br⁻ oxidized to Br₂.*”

Sub3: The measurement of the volume fraction of nucleation and growth is problematic due to the large spot volume. As the determination of NC concentration profiles is key, the authors should explain how they manage to obtain this profile in a seemingly quantitative way.

Response: We apologize for the confusion. The spot of beam we are using is 34 μm, which is much narrower than typical synchrotron X-ray beams (which are normally in the range of mm), and therefore allowed us to accurately probe volume fraction with a step size of 50 μm. All the data shown in this paper has a spatial resolution of 50 μm, taking Figure 4a as an example. In our measurement, we find that the volume fraction near the anode rises so fast that the 34 μm beam is too wide to probe the solution in equilibrium with the superlattice (which is ~1 μm thick). To make this point clear, we modified our text to “*It is worth noting that our experimental setup allows a spatial resolution of 50 μm, and the reported local volume fraction is a value averaged over this width. Figure 4a suggests that the volume fraction rises ~10 times within the 250 μm slice of solution next to the anode and it rises faster for the solution closest to the anode. For this reason, we expect a significant volume fraction rise within the first 50 μm slice of solution. As a result, the volume fraction of nanocrystal solution in equilibrium with superlattices, which are 1-2 μm thick, is expected to be much higher than the reported average value.*”

Sub4: Remarkably, the lowest fields give the largest superlattice domains, and the best quality crystals according to the structure factor profile, figure 6 f. Please comment, especially the paradox that with the lowest fields, the polydispersity in the volume where the superlattices grow is the largest.

Response: Weak fields generate smaller nanocrystal flux, which in turn leads to a lower nucleation density and larger average domain size. In our opinion, there are two factors contributing to the fact that the lowest field leads to best quality crystal: (1) the crystal

grows at the slowest rate under the lowest field, which is closest to the equilibrium condition and produces the fewest defects; and (2) the lower field is more efficient at size-selection. Electric fields, regardless of strength tend to accelerate larger nanocrystals to a higher velocity, and therefore preferentially accumulate larger nanocrystals at the anode. The non-intuitive effects come about because we have a finite population of nanocrystals. A weaker field drives a smaller proportion of the total nanocrystal population to the electrode – namely, the proportion with the largest size; whereas stronger fields drive a larger proportion of the total nanocrystal population to the electrode, which will necessarily include a broader distribution of the largest nanocrystals (i.e the largest size and also those with not as large a size) leading to a higher polydispersity and consequently poorer-quality crystals. At an imaginary extreme limit, a very strong electric field drives the entire nanocrystal population to the anode, resulting in no size-selection effect.

Sub5: It is stated that larger fields favor the drift of the larger NCs to the growth area. This seems to be in contradiction with the fact that the fcc lattice constant is the smallest for the strongest fields (Figure 6g). Please explain.

Response: This is part/continuation of the previous comment, and we apologize for the confusion. The message we would like to deliver is not “larger fields favor the drift of the larger NCs” but all fields favor the drift of larger NCs. Larger fields accumulate a larger proportion of the nanocrystal population, which therefore will have an average size closer to the starting bulk population. In other words, the effect of selecting large nanocrystals is less pronounced for stronger fields. We appreciate that this is counter-intuitive, and we modified our description in the text to clarify our idea “*For all the studied field strengths, the average nanocrystal size at the anode surface are larger than the initial value prior to applying a field. The size increase, however, is smaller for a stronger field. This is because there are a finite number of nanocrystals in the system. Stronger fields accumulate more nanocrystals to the anode, which represent a larger proportion of the total population and therefore have an average size closer to the starting population, resulting in a smaller change. In other words, the effect of “selecting large nanocrystals” is less pronounced for stronger fields.*” This is noteworthy because most previous EPD studies of nanocrystal deposition have used fields several orders of magnitude larger than those used in this study where size-selection is less effective.

REVIEWERS' COMMENTS:

Reviewer #2 (Remarks to the Author):

I am ok with the modifications the authors made and support the publication of the paper in its current form.

Reviewer #3 (Remarks to the Author):

I agree with the Response and the changes made.